# Physicians’ Perspectives on a Multi-Dimensional Model for the Roles of Electronic Health Records in Approaching a Proper Differential Diagnosis

**DOI:** 10.3390/jpm13040680

**Published:** 2023-04-19

**Authors:** Abdullah Alanazi, Amal Almutib, Bakheet Aldosari

**Affiliations:** 1Health Informatics Department, King Saud Ibn Abdulaziz University for Health Sciences, Riyadh 11481, Saudi Arabia; 2King Abdullah International Medical Research Center, Riyadh 14611, Saudi Arabia

**Keywords:** electronic health record, differential diagnosis, physician, information system

## Abstract

Many healthcare organizations have adopted Electronic Health Records (EHRs) to improve the quality of care and help physicians make proper clinical decisions. The vital roles of EHRs can support the accuracy of diagnosis, suggest, and rationalize the provided care to patients. This study aims to understand the roles of EHRs in approaching proper differential diagnosis and optimizing patient safety. This study utilized a cross-sectional survey-based descriptive research design to assess physicians’ perceptions of the roles of EHRs on diagnosis quality and safety. Physicians working in tertiary hospitals in Saudi Arabia were surveyed. Three hundred and fifty-one participants were included in the study, of which 61% were male. The main participants were family/general practice (22%), medicine, general (14%), and OB/GYN (12%). Overall, 66% of the participants ranked themselves as IT competent, most of the participants underwent IT self-guided learning, and 65% of the participants always used the system. The results generally reveal positive physicians’ perceptions toward the roles of the EHR system on diagnosis quality and safety. There was a statistically significant relationship between user characteristics and the roles of the EHR by enhancing access to care, patient–physician encounter, clinical reasoning, diagnostic testing and consultation, follow-up, and diagnostic safety functionality. The study participants demonstrate positive perceptions of physicians toward the roles of the EHR system in approaching differential diagnosis. Yet, areas of improvement in the design and using EHRs are emphasized.

## 1. Introduction

Electronic Health Records (EHRs) play a critical role in enhancing the quality of healthcare services and improving patient safety [1]. The EHR ensures continuity of care through its ability to integrate patient information among and across different departments and care settings and providers [2].

EHR systems collect patient information such as demographic data, medical history, compliant issues, diagnosis, progress notes, types of treatments and medication, vital signs, lab results, radiology reporting, and imaging [3]. This information is available whenever the provider asks—and it is also used to alert care providers for any medication errors, critical lab results, and types of allergies to be observed [4]. In addition, it enables some secondary use objectives such as clinical and health research, health management, and quality assurance [5]. The potential of EHRs in enhancing the quality of clinical decisions is evident, as access data in EHRs are correlated with improving diagnostic accuracy when compared to no access to EHRs [6]. Furthermore, EHRs help properly utilize diagnostic tests and develop computer-aided diagnostic (CAD) tools by compiling clinical notes, laboratory results, and radiological images [7,8].

In addition to improving the quality of the diagnosis process and its outcomes, it facilitates clinical documentation and, thus, consequently, care coordination and continuity through maintaining updated problem lists, tests, and medications [9]. As a result, we can develop approaches and tools to improve clinical reasoning by incorporating decision-support functionality to create an appropriate differential diagnosis. Thus, designing an EHR system that meets the needs of good care quality is critical [10]. This was attainable in the early detection of heart failure when analyzing clinical notes of primary care providers based on diagnostics criteria for heart failure [11]. Another study applied proper diagnostics criteria that helped detect delayed prostate and colorectal cancer (CRC) cases [12]. Assessing missed opportunities to avoid unnecessary hospitalization by applying proper diagnostics criteria in the clinical notes of the primary care providers for patients who visited primary care before admission has yielded that 21% of hospitalizations were avoidable [13]. Furthermore, EHRs can provide communication portals to patients at home and support telehealth, and mobile e-health applications would support patient engagement.

On the other hand, the diagnosis-making process is becoming more difficult considering the impaired face-to-face communication in the EHR environment, reliance on the paging system for consultation, and the overwhelming and unstandardized data the EHR brings [14]. Even with the availability of physical measures, there is a need for more certainty in diagnosis. The inability to accurately explain the patient’s health problem is obvious in discussing diagnosing obesity, psychiatric, and many other diseases [15]. Speaking of data fragmentation, the annual Catalan government health survey reveals more comorbidity among the population than the comorbidity based on the data found in their EHR (60% vs. 40%) [16]. Moreover, the EHR brought changes in healthcare workflow, requiring care process redesign, system maintenance, and upgrades. Furthermore, it comes with unintended consequences, such as new errors (ex., juxtaposition) and changes in power and the communication pattern. This comes with other types of risk; in one study, IT vulnerabilities were responsible for 248 cases (<1%) of medical malpractice claims, which were related to medication (31%) and diagnosis (28%) [17]. Therefore, more studies should be conducted to assess the physicians’ requirements in the EHR system to serve the diagnosis process and assess the role of EHRs in approaching accurate diagnosis.

The study aims at surveying physicians’ perspectives on the roles of EHRs in approaching a proper differential diagnosis by assessing the impact of EHRs on access to care, patient–physician encounter, clinical reasoning, diagnostic testing and consultation, follow-up, and diagnosis safety functionality.

## 2. Materials and Methods

This is cross-sectional descriptive research using a survey-based approach. The study was conducted on physicians using Electronic Health Records in Saudi Arabia. Considering that the study population size is 43,000 (total number of physicians in Saudi Arabia) and the confidence level of 95% and 5% as a margin of error, the recommended calculated sample size was 381 subjects. The validated survey gathered information about the important aspects of making differential diagnosis. The model of the study was adapted from the Graber et al., 2017 study [18]. Below is Table 1, which explains the dimensions and attributes of the study. The wording of the statements was modified and updated to accommodate the current EHR functions. The survey has a cover page that provides a brief statement about the study’s purpose and assurance of the respondents’ confidentiality and privacy. The first section (Questions 1–6) gathered the demographic and background information of the participant, including Gender, Age, Healthcare Experience, and Area of Work. The second section (Questions 1–3) was about Computer and IT Knowledge and Experience. The third section of the questionnaire contains 31 statements that cover the six dimensions of the study: access to care, patient–physician encounter, clinical reasoning, diagnostic testing and consultation, follow-up, and diagnostic safety functionality (Table 1). The survey was in a statement style. Responses were recorded on a five-stage Likert scale (Strongly Agree, Somewhat Agree, Neutral, Disagree, and Strongly Disagree) with a numeric score corresponding to each response. Three experts revised and validated this survey for face validity and to ensure the questionnaire was valid to assess the study’s objectives.

## 3. Results

Four hundred and nineteen questionnaires were distributed, and three hundred and ninety-four were returned and analyzed for data integrity and accuracy. Forty-three surveys were excluded due to missing information or response bias (i.e., choosing neutral to all answers). After reviewing the questionnaires, the total number of surveys included in the analysis was three hundred and fifty-one, representing 83.8% of the total amount distributed. Cronbach’s alpha test was used to check for the reliability and internal consistency of the study variables. The Cronbach alpha scores for the study variables were as follows: access to care (0.807), patient–physician encounter (0.774), clinical reasoning (0.873), diagnostic testing and consultation (0.805), follow-up (0.825), and diagnostic safety functionality (0.915) (Table 2). The reliability scores were acceptable to continue for further analysis.

### 3.1. Sample Characteristics

Three hundred and fifty-one participants were included in the study, of which 61% were male. Most of the participants were 30–39 years of age (42%), 37% of the participants were from the Middle East, and 35% were Saudis. Regarding healthcare experience, the majority of the participants had 10–14 years of experience (42%). The main group of participants was from family/general practice (22%), medicine, general (14%), and OB/GYN (12%). The participants were registrar (44%), which was followed by residents (24%), and consultants (22%). Overall, 66% of the participants ranked themselves as IT competent (66%) and proficient users (18%). Regarding IT training, most participants underwent IT self-guided learning (66%), and 21% went through formal training. Regarding system use, 65% of the participants always used the system, while 18% revealed their usage as sometimes. Table 3 shows the demographics and the characteristics of the study participants.

### 3.2. Responses on the Dimensions of the Study Model

The participants show a high level of agreement on the positive roles of EHRs in approaching differential diagnosis. The average mean of participants’ responses for each dimension is between 3.72 and 4.22, corresponding to the five-point Likert scale. The role of the EHR in the clinical reasoning dimension reveals the highest agreement (4.22), which was followed by diagnostic testing and consultation (4.21), patient–physician encounter (4.08), and follow-up (4.07). Table 4 demonstrates the frequency of responses on the dimensions of the study model and the means and standard deviations (SD) of their responses. Additionally, the highest agreed-on attributes (4 and above on the Likert scale) within each dimension are presented in sequential order in Table 5.

### 3.3. Differences in Means of Response to Study Dimensions Based on Participants’ Characteristics

Assessing the differences in responses to study dimensions based on participants’ profiles reveals a statistically significant difference in responses to the role of EHRs in clinical reasoning (t = 2.278, sig < 0.05) based on gender in preference to the female. While other participants’ characteristics and study dimensions (access to care, patient–physician encounter, clinical reasoning, diagnostic testing and consultation, follow-up, and diagnostic safety functionality) were assessed through ANOVA analysis.

As work experience increases, the participants tend to appreciate the roles of her in approaching accurate differential diagnosis across all study dimensions (sig < 0.05). Younger participants appreciated the roles of EHRs in enhancing the quality of diagnosis by providing means to access care, diagnostic safety functionality, and follow-up compared to older participants (f = 12.7, f = 3.3, f = 5.03, respectively, with sig < 0.05). Competent and proficient IT participants and frequent EHR users appreciated the role of the EHR in providing better access to care compared to their counterparts (f = 4.4, with sig < 0.05). Those who have received formal training have more positive perceptions toward the roles of the EHR in enhancing clinical reasoning, ability to access care, and impact on patient–physician encounters compared to those who received other types of training (f = 4.01, f = 3.46, f = 3.4, respectively, with sig < 0.05).

## 4. Discussion

Our study assesses the ability to approach an accurate diagnosis using EHRs to improve the six dimensions contributing to the diagnosis-making process, consequently improving care services, diagnostic functionalities, and patient safety. Diagnostic errors were found in 15% of patient medical records upon reviewing patients at risk in one study [19]. Furthermore, patients found a mistake in 1 of every five charts when asked to review their records; 40% of these errors were perceived as serious and mostly related to diagnostic errors [20]. However, there is a debate over the validity of reviewing patient charts to detect diagnostic errors [19]. Furthermore, Singh and Sitting, in their study, have called to advance the science of measuring diagnostic mistakes by developing a multifaceted framework (The Safer Dx framework) to monitor and improve diagnostic errors [21].

Generally, participants of the study showed favorable roles of the EHR in improving their diagnostics accuracy. This can contribute to having accurate, prompt diagnosis data in the EHR system [22]. In addition, having structured data in the EHR system can also help ensure the completeness and accuracy of the captured data, enhancing diagnostic accuracy [23].

Regarding the assessed dimensions of the study, the role of electronic health records in clinical reasoning was the most appreciated dimension to explain the role of the EHR in approaching differential diagnosis. This goes along with the study of Wills et al. in valuing the impact of EHRs on clinical reasoning [24]. Exchanging the data to serve the inter- and multi-disciplinary care approach has helped clinical reasoning for a proper diagnosis [10]. Furthermore, the participants acknowledged the importance of organizing and presenting the data at the point of care through the capacity of the EHR to capture, retrieve, and read clinical notes electronically at the point of care, and this would enhance clinical reasoning. Colicchio and Cimino conducted a systematic review in 2019 and revealed that clinicians’ reasoning is impacted by their ability to capture and use data at the point of care [25]. Furthermore, the ability to organize the data in the EHR system to know the current patient status would decrease the clinicians’ cognitive workload and enhance their ability for clinical reasoning [26]. In addition to patient data interconnectivity, the participants acknowledged the importance of allowing free text entry with a proper abstracting method aided by utilizing natural language processing (NLP), which would add to the value of the EHR [27]. Additionally, the participants reveal that using visualization techniques to display time-based data graphically and provide a 360 view of patient data would contribute to knowledge discovery and help in approaching differential diagnosis [28]. The participants recognized the importance of EHRs in supporting test-result communication. One study disclosed that 55% of physicians were dissatisfied with the current approach for notifying test results and requesting adding extra features for results tracking and management [29,30]. In addition, patient satisfaction can be improved additionally with an automated test results management system [31]. The participants expect that the EHR would support them by generating a reminders list. This feature is important, as one study showed that the EHR system could enhance the quality of care by providing point-of-care reminders and more prompt and valid feedback from clinicians, as it contributed to enhancing the quality measures for 16 chronic diseases and preventive care (14 measures were improved significantly) [30]. Nevertheless, the participants emphasized the need to design and implement these features for the overall safety of the diagnosis process regarding properly dealing with information overload and minimizing inappropriate alerts. For example, tailoring alerts for high-risk patients and possible serious adverse events would minimize alert fatigue and enhance the safety of diagnosis [32]. Further to the reminders, the participants appreciated the availability of point-of-care technologies that would allow enhancing the accessibility and safety of healthcare services [33]. Another important feature mentioned by the participants of the study is that the EHR should provide users with access to external knowledge resources. This would help physicians in answering questions and consequently become associated with the care decisions. A study revealed that there is on average 0.57 questions per patient; in 51% of cases, physicians were looking for answers and 78% of the questions were answered through accessing external knowledge resources [34]. Further to the clinical questions, the decision support system within the EHR can help in improving the problem list, which has an important role in facilitating continuity of care across different care providers and settings [35]. Artificial intelligence (AI) has the potential to mimic human thinking by designing specific algorithms to help in supporting the decision-making process. Therefore, clinicians can benefit from the applications of AI in different healthcare processes. One study reveals that utilizing AI in diagnosis resulted in an accuracy of 72.52%, which is comparable to the diagnosis accuracy rate of an average clinician (71.4%) [36]. Similarly, using Babylon GP is an AI application used in the UK as part of the NHS primary care services. A study found that the diagnosis accuracy of Babylon GP is equal to human primary care providers [37]. Specific algorithms can be designed to carry out specific tasks, such as analyzing patients’ historical data in line with the lab results and coming up with a specific diagnosis [38]. The results of using AL can even surpass the capacity of human doctors in some diseases; a study comparing human GP diagnoses of headaches and AI revealed that AI produces a more accurate diagnosis [39]. Another example is using AI for the early detection of cancer in the primary care setting, and this study reveals a promising result [40]. However, the performance of AI applications depends on the accuracy, robustness, and clinical plausibility of the algorithms [41]. Therefore, applying the said algorithms to the data of a specific hospital for a specific care process or function (such as diagnosis) may be influenced by the setting, quality of data, and the structure of the EHR system, and thus, the generalizability of the performance is questionable [42]. For instance, one study evaluated data accuracy and revealed that only 57% of the data are complete, and using the data across hospitals revealed that only 23% of instant data were comparable. Due to the diversity in the data quality attributes, drawing and generalizing specific conclusions from a particular case is challenging [42]. Thus, reaching findings would require a rigorous methodology to assess the quality of data in addition to removing noise and biases before running the desired algorithm [43]. Additionally, the causality of the findings is difficult to establish, especially since most of the findings were produced from descriptive or observational studies [44]. On the other hand, the current diagnosis process is practiced through an individual mode (physician only), which is prone to human errors and results in 10% errors in all diagnosis cases [45]. The National Academy of Medicine (NAM) proposed a new approach to diagnosis in which the process should be conducted in a team-based approach [46].

The findings of our study are compatible with other studies on how age impacts the role of EHRs in reaching an accurate differential diagnosis. The study of Chen and others indicates that physicians under 50 have more intention to use EHRs to promote the reliability of the diagnostic evaluation. Younger users usually have previous experience with computers, making them more accepting of using EHRs [47]. Moreover, the results of this study showed that competent users of information technology tend to appreciate the role of the EHR in differential diagnosis. There was a statistically significant difference between proficient users and beginners, in which skilled users would perform better in using the EHR system. This demonstrates the importance of IT knowledge and training for the practitioner to achieve the highest level of realization and enhance the quality of the differential diagnosis using the EHR system [48]. Regarding work experience, our findings demonstrated that the average years of experience (10 to 19 years) tend to appreciate the role of EHRs in enhancing the quality of diagnosis in all six dimensions of our study. Similarly, the findings of the study of Ramnarayan et al. displayed that physicians with more clinical experience tend to use clinical systems to enhance the quality of their diagnosis [49]. Compared to our findings for the impact of the EHR on patient–physician encounters, the study by Matthew mentioned that experienced users were more confident in allowing the patient access to their records. They are more likely to agree to use tools that could improve their relationship with the patient [50]. Moreover, our findings suggest that the importance of clinical reasoning and diagnostic testing in EHRs was more statistically significant for female physicians. This may be attributed to the findings of one study that indicates that females tend to have a sense of responsibility and over-check their patient’s results [51]. The said study found that compared to male doctors, female doctors had significant differences in drug–drug interaction checking and tended to have a better quality of care [51,52]. In general, the EHR system has enhanced communication between patients and physicians, allowed patients to be more active in their treatment, and increased self-efficiency and patient satisfaction [53]. Additionally, the decision-making process has improved patient outcomes by alerting them of possible adverse interactions [54].

Nevertheless, the current study has some limitations, as the target population included EHR physician users working in Saudi Arabia, and this may limit the generalizability of our findings. In addition, studying the user’s subjectivity via qualitative analysis when dealing with the system can add more depth and demonstrative findings than just using a quantitative way. A combination of interviews and a quantitative approach or even conducting an observational study can provide more meaningful results that can capture most of the users’ perceptions of the EHR system and lead to a more holistic approach that cannot be obtained using a quantitative way alone. In summary, the current research results demonstrated the success of the study’s impact variables in examining the effect of the EHR on approaching differential diagnosis. Physicians had a positive view of EHR roles in their daily work.

## 5. Conclusions

Physicians have positive perceptions toward the roles of the EHR in approaching differential diagnosis. The assessed dimensions of the study (access to care, patient–physician encounter, clinical reasoning, diagnostic testing and consultation, follow-up, and diagnostic safety functionality) have proved to be indicators to examine the role of the EHR in the diagnostic process in clinical practice. The findings of this study contribute to an assessment of the EHR system and can help future improvements and modifications regarding users’ requirements to serve the diagnosis process.

## Figures and Tables

**Table 1 jpm-13-00680-t001:** The dimensions and attributes of the study.

Dimension	Scope of the Dimension
Access to Care	Covers aspects related to setting communication portals with patients, telemedicine, and mobile applications and supporting open notes.
Patient–Physician Encounter	Covers aspects related to supporting patient engagement in the encounter and team-based diagnosis; it also has a role in improving ways to capture documentation.
Clinical Reasoning	Covers aspects related to organizing data and presenting it optimally at the point of care, incorporating decision support functionality to aid in calculations, allowing free text entry to document clinical reasoning, providing access to relevant medical knowledge at the point of care, and facilitating data searching and improving the problem list by using decision support.
Diagnostic Testing and Consultation	Covers aspects related to providing decision support for the appropriate selection of diagnostic tests, facilitating communication to appropriate expertise at the point of care (consultants, etc.), displaying time-based data graphically (lab tests, medications utilization, etc.) and about other clinical information, and utilizing computer-aided diagnostic algorithms to improve the detection and classification of X-rays and other visual data.
Follow-Up	Covers aspects related to supporting clinicians by being able to generate their reminder list, supporting test-result communication, registries and reminders to identify selected patients, and automating feedback on changes in diagnosis.
Diagnostic Safety Functionality	Covers aspects related to facilitating the use of trigger tools to identify patients at risk, supporting interoperability so that all relevant medical information can be gathered and used, discouraging designation of a diagnosis prematurely (e.g., NYD = not yet diagnosed), monitoring diagnostic performance (timeliness, accuracy), supporting high-quality documentation, considering how to combat information overload; minimizing inappropriate alerts, developing predictive analytic approaches to suggest likely diagnoses not considered.

**Table 2 jpm-13-00680-t002:** The coefficient value of Cronbach’s alpha.

N	Variables	Cronbach’s Alpha	Number of Items
1	Access to care	0.807	3
2	Patient–Physician encounter	0.774	5
3	Clinical reasoning	0.873	7
4	Diagnostic testing and consultation	0.805	4
5	Follow-up	0.825	4
6	Diagnostic safety functionality	0.915	8

**Table 3 jpm-13-00680-t003:** Participants’ demographic data.

Characteristic	Category	Percent
Gender	Male	61%
	Female	39%
Age	20–29	28%
	30–39	42%
	40–49	26%
	50 and above	4%
Nationality	Saudi	35.3%
	Western countries	7.7%
	Middle East	37.3%
	Other	19.7%
Healthcare Experience	<5	22%
	5–9	11%
	10–14	42%
	15–19	12%
	20 and above	13%
Work Specialized	Pediatrics	11%
	Medicine, general	14%
	OB/GYN	12%
	Family/general practice	22%
	Other	19%
Professional Rank	Consultant	22%
	Registrar	45%
	Residents	24%
	Others	13%
IT Skills	Advanced beginner	12%
	Competent	66%
	Proficient	18%
	Expert	4%
IT Training	Formal training	27%
	Workshops	17%
	Self-guided learning	44%
	No training	12%
IT Use	Always	75%
	Sometimes	18%
	Rarely	7%

**Table 4 jpm-13-00680-t004:** Frequency of responses on the dimensions of the study model.

Dimension	S. Disagree	Disagree	Neutral	Agree	S. Agree	Mean	SD
	% Participants					5 Pts Likert Scale	
Access to Care	4.3	7	28.7	31.7	28.3	3.72	0.22
Patient–Physician Encounter	0	2.6	19.4	45	33	4.08	0.28
Clinical Reasoning	0.3	1	18.3	36.8	43.6	4.22	0.42
Diagnostic Testing and Consultation	0	2.7	20.5	29.3	47.5	4.21	0.27
Follow-Up	0.5	1.2	24.3	38.5	35.5	4.07	0.27
Diagnostic Safety Functionality	2.5	2	31.1	37.5	26.8	3.84	0.58

**Table 5 jpm-13-00680-t005:** The most agreed-on attributes within each dimension.

Dimension	Attributes
Clinical Reasoning	Organizes data and present it optimally at the point of care
	Provides access to relevant medical knowledge at the point of care
	Allows free text entry to document clinical reasoning
	Improves the problem list by using decision support
	Incorporates decision support functionality to aid in calculations
Diagnostic Testing and Consultation	Display time-based data graphically (lab tests, medication utilization, etc.)
	Provide decision support for the appropriate selection of diagnostic tests
	Facilitate communication to appropriate expertise at the point of care (consultants, etc.)
Patient–Physician Encounter	Improves ways to capture documentation.
	Supports team-based diagnosis
	Supports patient engagement in the encounter
Follow-Up	Support test-result communication
	Support clinicians by being able to generate their reminder list
Diagnostic Safety Functionality	Consider how to combat information overload; minimize inappropriate alerts
	Develop predictive analytic approaches to suggest likely diagnoses not considered
	Facilitate the use of trigger tools to identify patients at risk
Access to Care	Supports telemedicine and mobile applications

## Data Availability

Not applicable.

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
