# Peer review of "Physicians’ Perspectives on a Multi-Dimensional Model for the Roles of Electronic Health Records in Approaching a Proper Differential Diagnosis"

_jpm, 2023, doi:10.3390/jpm13040680_

Round 1

Reviewer 1 Report

Review of "Physicians' Perspectives on a multi-dimensional model for the Roles of Electronic Health Records in approaching a Proper Differential Diagnosis."

The authors intended to survey the physicians' perspectives on the roles of EHR in approaching a proper differential diagnosis by assessing the impact of EHR on multiple dimensions. The authors found a statistically significant relationship between user characteristics and the roles of the EHR. The research question is indeed important and the betterment of EHR database guarantees improved diagnosis of diseases. I don't have any major complaints except for a few minor and structural ones, which I have listed below.

Comments:

1.     Page 2, line 86. The citation [10] is placed after the period. I think it was intended to be before the period.

2.     Table 2. Extra space between words in “Diagnostic  testing and consultation” and “Diagnostic     safety functionality” in the dimension column.

3.     Table 2. Weird alignment in the right column corresponding to “Clinical Reasoning”.

4.     Section 3. Why did 419 questionnaires were collected if the necessary sample size calculated was 381?

5.     Table 3. What is “IS use”? I think it was intended to be “IT use”.

6.     Page 5, line 129. Is it a heading for a subsection? The similar headings can also be found after Tables 2 and 5. If it is a heading, please use bold face or italics.

7.     Table 4. The formatting of this table (and the other tables too) can be improved. If abbreviating the word “Strongly”, please do that for all places or don’t do it at all. Moreover, having a top and bottom line of a table better separates the table from the text. The rest of the horizontal and vertical lines can be adequately used to separate different rows/sections/columns.

8.     Table 6. Personally I don’t feel this table is necessary. On the other hand, a series of grouped boxplots (for different user characteristics) better depicts the overall picture. The authors can include the p-value on the boxplots to show if the differences are significant or not.

9.     If possible, the authors can provide the complete questionnaire in the appendix or a supplement.

Author Response

Dear Reviewer, 

Thank you for your valuable points. I tried to address and definitely it has enriched our paper. 

Thank you, 

Reviewer 2 Report

Did you encounter any significant issues in both forward and backward translation? Did you perform any pilot testing?

Who assessed participants?

What was the definition of IT Skills, and what were the criteria for classifying participants into four groups?

Your sample size calculation was 381 participants but the study included 351 participants, why?

It would be useful if you could describe the statistical methodology.

Author Response

(The authors gave the same response as above.)
